# 3D Indoor Position Estimation Based on a UDU Factorization Extended Kalman Filter Structure Using Beacon Distance and Inertial Measurement Unit Data [note 1]

**DOI:** 10.3390/s24103048

**Published:** 2024-05-11

**Authors:** Tolga Bodrumlu, Fikret Caliskan

**Affiliations:** 1Mechatronics Engineering Department, Istanbul Technical University, Istanbul 34025, Turkey; 2Control and Automation Engineering Department, Istanbul Technical University, Istanbul 34025, Turkey; caliskan@itu.edu.tr

**Keywords:** indoor positioning, extended Kalman filter, UDU factorization, sensor fusion

## Abstract

The development of the GPS (Global Positioning System) and related advances have made it possible to conceive of an outdoor positioning system with great accuracy; however, for indoor positioning, more efficient, reliable, and cost-effective technology is required. There are a variety of techniques utilized for indoor positioning, such as those that are Wi-Fi, Bluetooth, infrared, ultrasound, magnetic, and visual-marker-based. This work aims to design an accurate position estimation algorithm by combining raw distance data from ultrasonic sensors (Marvelmind Beacon) and acceleration data from an inertial measurement unit (IMU), utilizing the extended Kalman filter (EKF) with UDU factorization (expressed as the product of a triangular, a diagonal, and the transpose of the triangular matrix) approach. Initially, a position estimate is calculated through the use of a recursive least squares (RLS) method with a trilateration algorithm, utilizing raw distance data. This solution is then combined with acceleration data collected from the Marvelmind sensor, resulting in a position solution akin to that of the GPS. The data were initially collected via the ROS (Robot Operating System) platform and then via the Pixhawk development card, with tests conducted using a combination of four fixed and one moving Marvelmind sensors, as well as three fixed and one moving sensors. The designed algorithm is found to produce accurate results for position estimation, and is subsequently implemented on an embedded development card (Pixhawk). The tests showed that the designed algorithm gives accurate results with centimeter precision. Furthermore, test results have shown that the UDU-EKF structure integrated into the embedded system is faster than the classical EKF.

## 1. Introduction

With the advancement of technology in recent years, positioning systems have advanced significantly; however, these systems typically take place outdoors, where GPS signals are readily available. For indoor environments, GPS signals are insufficient for positioning accuracy because they are not at the desired level. This situation creates problems for indoor unmanned aerial (UAVs) and ground vehicles (UGVs). Therefore, studies have begun on position estimation for indoor environments. There are lots of different approaches related to indoor positioning, such as those that are motion-captured-based, lidar-based, Wi-Fi-based, Bluetooth-based, ultra-wideband (UWB)-based, and IMU-based [1,2,3]. Motion-capturing-based systems like VICON and OptiTrack employ multiple high-speed cameras to determine the position of an object relative to one another; however, these systems have a drawback in that they require a complex setup and challenging calibration. Lidar-based positioning approaches, such as Gmapping, Hector, and Cartographer [4,5,6], are also an option, but they only provide the two-dimensional position due to weight limitations, especially for UAVs.

Wi-Fi- and Bluetooth-based positioning techniques are widely used, with an accuracy of a few meters [7]; however, they have constraints regarding the required number of access points, costs, and energy consumption. Bluetooth Low Energy (BLE) and Wi-Fi operate on the same frequency. BLE is intended for short-range and energy-efficient communication using brief messages [8]. BLE-based location is usually achieved by placing proximity beacons at specific locations. Receivers determine their position by measuring the RSSI (receiver’s distance from the sender) of the closest beacons.

UWB positioning is lightweight, has a straightforward design, provides stable positioning results, and can achieve a centimeter level of accuracy; however, relying solely on UWB does not fulfill the demands of high-precision indoor operations. Although an IMU is a popular sensor for determining orientation, its position estimates can accumulate errors over time due to drift [9,10].

Therefore, an algorithm is designed to perform more accurate indoor positioning estimation. This paper presents an algorithm that uses the EKF with UDU factorization to combine the data from an IMU and raw distance data to calculate an accurate position estimation. The algorithm begins by using the RLS method with a trilateration algorithm to obtain an initial position estimation. This estimation is then fused with data from the IMU’s acceleration sensor.

The main contributions of this work are listed below:The use of a sensor fusion algorithm, utilizing the EKF structure, in conjunction with acceleration data from IMU sensors, to enhance the accuracy of position information obtained from distance data provided by beacon sensors via the RLS algorithm.The incorporation of a UDU factorization structure to reduce computation costs in embedded systems, in addition to the utilization of the EKF structure for sensor fusion.The ability of the designed algorithm to produce a solution even under suboptimal conditions, such as the use of only three beacon sensors instead of the ideal four.

In this article, detailed information will be provided on the position estimation algorithm, followed by a description of the data collection process from the real sensor, and, finally, the results of the real-time tests of the designed algorithm will be conveyed.

## 2. Position Estimation Algorithm

In this section, the structure of algorithms designed for position estimation will be discussed. Figure 1 illustrates the general framework of the algorithm. Initially, information related to the geometric approach will be provided, and, subsequently, detailed information about the sensor fusion aspect will be conveyed. The designed algorithm is intended to be applied in real-time systems. For this reason, the EKF structure has been decomposed into UDU factorization to reduce the computation cost. Hence, the UDU-EKF structure is employed instead of the classical EKF structure in this study.

### 2.1. Geometric Approach

#### 2.1.1. Solution Based on Three Reference Points

This study is based on a geometric method, as depicted in Figure 2, which involves three reference points, *B*_1_, *B*_2_, and *B*_3_, represented by their coordinates: (*x*_1_, *y*_1_, *z*_1_), (*x*_2_, *y*_2_, *z*_2_), and (*x*_3_, *y*_3_, *z*_3_), respectively. Additionally, the study also includes distance measurements, *d*_1_, *d*_2_, and *d*_3_, up to point A. The determination of the coordinates (*x*, *y*, *z*) of point A is equivalent to finding the solutions to the following system of quadratic equations [11,12,13].
(1)$$\begin{array}{c} {\left(x-x_{1}\right)}^{2}+{\left(y-y_{1}\right)}^{2}+{\left(z-z_{1}\right)}^{2}={d_{1}}^{2}\\ {\left(x-x_{2}\right)}^{2}+{\left(y-y_{2}\right)}^{2}+{\left(z-z_{2}\right)}^{2}={d_{2}}^{2}\\ {\left(x-x_{3}\right)}^{2}+{\left(y-y_{3}\right)}^{2}+{\left(z-z_{3}\right)}^{2}={d_{3}}^{2} \end{array}$$

The equations in this system can be expressed as follows [11,12,13]:(2)$$\begin{array}{c} {(x}^{2}+y^{2}+z^{2})-2x_{1}x-2y_{1}y-2z_{1}z=d_{1}^{2}-x_{1}^{2}-y_{1}^{2}-z_{1}^{2}\\ {(x}^{2}+y^{2}+z^{2})-2x_{2}x-2y_{2}y-2z_{2}z=d_{2}^{2}-x_{2}^{2}-y_{2}^{2}-z_{2}^{2}\\ {(x}^{2}+y^{2}+z^{2})-2x_{3}x-2y_{3}y-2z_{3}z=d_{3}^{2}-x_{3}^{2}-y_{3}^{2}-z_{3}^{2} \end{array}$$

The system of equations can be represented in matrix form as follows [11,12,13]:(3)$$A_{0}u=b_{0}$$
where
(4)$$A_{0}=\left[\begin{array}{cccc} 1 & {-2x}_{1} & {-2y}_{1} & {-2z}_{1}\\ 1 & {-2x}_{2} & {-2y}_{2} & {-2z}_{2}\\ 1 & {-2x}_{3} & {-2y}_{3} & {-2z}_{3} \end{array}\right],u=\left[\begin{array}{c} x^{2}+y^{2}+z^{2}\\ x\\ y\\ z \end{array}\right],b_{0}=\left[\begin{array}{c} d_{1}^{2}-x_{1}^{2}-y_{1}^{2}-z_{1}^{2}\\ d_{2}^{2}-x_{2}^{2}-y_{2}^{2}-z_{2}^{2}\\ d_{3}^{2}-x_{3}^{2}-y_{3}^{2}-z_{3}^{2} \end{array}\right]$$

The solution set of this system has two possible interpretations: In one interpretation, points *B*_1_, *B*_2_, and *B*_3_ are not in a straight line. In the other interpretation, they are in a straight line [11,12,13].

**Case 1. *B*_1_, *B*_2_, and *B*_3_ are not in a straight line**.

In this situation, the following propositions hold true:The rank of matrix *A*_0_ is 3.The dimension of the null space of *A*_0_ is 1.

The general solution to Equation (3) can be represented as follows:(5)$$u=u_{p}+t.u_{h}$$
where *t* is a real number coefficient, *u_p_* is a particular solution of (3), and *u_h_* is a solution of the homogenous system $A_{0}.u=0$. Both *u_p_* and *u_h_* can be calculated using the Gaussian elimination method:(6)$$u_{p}={\left(u_{p0,}u_{p1,}u_{p2,}u_{p3}\right)}^{T}\;,\;u_{h}={\left(u_{h0,}u_{h1,}u_{h2,}u_{h3}\right)}^{T},u={\left(u_{0,}u_{1,}u_{2,}u_{3}\right)}^{T}$$

If we substitute the expression for u_p_, u_h_, and u into Equation (4), the following expressions are obtained:(7)$$\begin{array}{c} u_{0}=u_{p0}+tu_{h0},\;u_{1}=u_{p1}+tu_{h1\;,\;}\\ u_{2}=u_{p2}+tu_{h2},\;u_{3}=u_{p3}+tu_{h3} \end{array}$$

By using the constraint *u ∈ E*:(8)$${u_{p0}+tu_{h0}=(u_{p1}+tu_{h1})}^{2}+{\left(u_{p2}+tu_{h2}\right)}^{2}+{\left(u_{p3}+tu_{h3}\right)}^{2}$$
and then
(9)$$t^{2}\left(u_{h1}^{2}+u_{h2}^{2}+u_{h3}^{2}\right)+u_{p1}^{2}+u_{p2}^{2}+u_{p3}^{2}-u_{p0}+t\left(2u_{p1}u_{h1}+2u_{p2}u_{h2}+2u_{p3}u_{h3}-u_{h0}\right)=0$$

This equation is a quadratic in the form of ${at}^{2}+bt+c=0$, and has two solutions:(10)$$t_{1/2}=\frac{-b\pm \sqrt{b^{2}-4ac}}{2a}$$

The solutions of the equation system can be presented as follows:(11)$$u_{1}=u_{p}+{t_{1}u}_{h},\;u_{2}=u_{p}+{t_{2}u}_{h}$$


**Case 2. *B*_1_, *B*_2_, and *B*_3_ are in a straight line.**


In this situation, the following propositions hold true:The rank of matrix *A*_0_ is 2.The dimension of the null space of *A*_0_ is 2.

The general solution to Equation (3) can be expressed as follows:(12)$${u=u}_{p}+t.u_{h1}+k.u_{\begin{array}{c} h2\\ \; \end{array}}$$
where *u_p_* is a particular solution of Equation (3), and *u_h_*_1_ as well as *u_h_*_2_ are two linearly independent solutions of the homogeneous system described by *A*_0_*.u =* 0.

#### 2.1.2. Solution Based on More Than Three Reference Points

If there is an additional distance of *d*_4_, *d*_5_, *… d_n_* to the reference points, *B*_4_, *B*_5_
*… B_n_* can be extend as follows [11,12,13]:(13)$$\left[\begin{array}{cccc} 1 & -2x_{1} & -2y_{1} & -2z_{1}\\ 1 & -2x_{2} & -2y_{2} & -2z_{2}\\ 1 & -2x_{3} & -2y_{3} & -2z_{3}\\ \vdots & \vdots & \vdots & \vdots \\ 1 & -2x_{n} & -2y_{n} & -2z_{n} \end{array}\right]\left[\begin{array}{c} x^{2}+y^{2}+z^{2}\\ x\\ y\\ z \end{array}\right]=\left[\begin{array}{c} {d_{1}}^{2}-{x_{1}}^{2}-{y_{1}}^{2}-{z_{1}}^{2}\\ {d_{2}}^{2}-{x_{2}}^{2}-{y_{2}}^{2}-{z_{2}}^{2}\\ {d_{3}}^{2}-{x_{3}}^{2}-{y_{3}}^{2}-{z_{3}}^{2}\\ \vdots \\ {d_{n}}^{2}-{x_{n}}^{2}-{y_{n}}^{2}-{z_{n}}^{2} \end{array}\right]$$

Expressed in a known form, this system can be written as follows:(14)$$\begin{array}{c} Au=b\\ A=\left[\begin{array}{cccc} 1 & -2x_{1} & -2y_{1} & -2z_{1}\\ 1 & -2x_{2} & -2y_{2} & -2z_{2}\\ 1 & -2x_{3} & -2y_{3} & -2z_{3}\\ \vdots & \vdots & \vdots & \vdots \\ 1 & -2x_{n} & -2y_{n} & -2z_{n} \end{array}\right],\;u=\left[\begin{array}{c} x^{2}+y^{2}+z^{2}\\ x\\ y\\ z \end{array}\right],b=\left[\begin{array}{c} {d_{1}}^{2}-{x_{1}}^{2}-{y_{1}}^{2}-{z_{1}}^{2}\\ {d_{2}}^{2}-{x_{2}}^{2}-{y_{2}}^{2}-{z_{2}}^{2}\\ {d_{3}}^{2}-{x_{3}}^{2}-{y_{3}}^{2}-{z_{3}}^{2}\\ \vdots \\ {d_{n}}^{2}-{x_{n}}^{2}-{y_{n}}^{2}-{z_{n}}^{2} \end{array}\right] \end{array}$$

The general solution, denoted as $\hat{u}$, can be obtained using the least squares method as follows:(15)$$\hat{u}={\left(A^{T}A\right)}^{-1}A^{T}b$$

The projection of point *p* on the column space of matrix *A* is as follows:(16)$$p={A(A^{T}A)}^{-1}A^{T}b$$

In this case, the coordinates of point *p* in the column space of Col (A) correspond to the solution, $\hat{u}$. Moreover, vector *b* consists of distances between the unknown point, *A*, and all the reference points; however, if the measurements are uncorrelated but have varying levels of uncertainty, the weighted least squares (WLS) method is utilized. The solution, $\hat{u}$, can then be determined using the following equation:(17)$$\hat{u}={\left(A^{T}V^{-1}A\right)}^{-1}A^{T}V^{-1}b$$

*V* is the covariance matrix of random errors. These errors refer to the variability or uncertainty in the measurements of the distances between points (represented by the vector *u*) and the reference points. It might be measurement noise (e.g., signal interference) or propagation effects (e.g., signal reflection or refraction). The RLS method uses *u*_0_ as the initial solution. It is continually updated with each new distance measurement, resulting in the creation of a new solution, *u*_1_. This approach enables the simultaneous calculation of both distance measurements and positioning, allowing for the initiation of position assignment even when not all distances are known, thus reducing waiting time and improving the speed of the positioning calculation. Generally, for indoor positioning, a limited number of participating reference stations does not cause a big challenge to the computational resources for multilateration. This presented algorithm utilizes a linear algebraic approach with low computational complexity, making it suitable for real-time applications.

The combination of distance data and the RLS algorithm is used to calculate the position. The following section will go into further detail on how to improve position estimation using the EKF.

### 2.2. Sensor Fusion Algorithm

Several algorithms are used for sensor fusion, including Bayesian networks, convolutional neural networks, Dempster–Shafer theory, and the Kalman filter [14]. This system employs the Kalman filter to achieve more precise position estimation; however, the Kalman filter is optimized for linear systems, while real-world systems often exhibit non-linearity. Thus, the EKF is utilized to enhance performance [15]. The implementation of the EKF used in this study will be discussed in further detail. The state vector is defined as follows [13]:(18)$$X[k]=\left[\begin{array}{c} p(w)[k]\\ v(w)[k]\\ a(w)[k] \end{array}\right]$$

The vector *p*(*w*) represents the coordinates of the object along the *x*, *y*, and *z* axes in the world coordinate system, and is given as [*p_x_*[k], *p_y_*[k], *p_z_*[k]]. The velocity vector of the object along the *x*, *y*, and *z* axes in the world coordinate system is represented by *v(w)* = [*v_x_*[k], *v_y_*[k], *v_z_*[k]]. Additionally, the accelerometer vector of the object in the world coordinate system is represented by *a(w)* = [*a_x_*[k], *a_y_*[k], *a_z_*[k]].

The world coordinate system here (*w*) can also be specified as the coordinate system of the UWB sensor (Marvelmind). At this point, ∆*t* is defined as the sampling time interval, and (∆*t*)*w*[*k*] can be defined as the process noise of acceleration, representing uncertainties in the acceleration process. $\frac{{∆t}^{2}}{2}w[k]$, $\frac{{∆t}^{3}}{6}w[k]$ can be represented as the process noise of velocity and process noise of positions, respectively. Equations of motion in the (*k* + 1) time interval can be expressed as follows [13]:(19)$$p[k+1]=p[k]+v[k]\left(∆t\right)+a[k]\frac{{∆t}^{2}}{2}+\frac{{∆t}^{3}}{6}w[k]$$

The equation states that the new position at time step *k* + 1 is determined by the current position, the velocity multiplied by the time step, half of the acceleration multiplied by the square of the time step, and a noise term. The noise term is added to account for uncertainties or external disturbances that may affect the motion of the system:(20)$$v[k+1]=v[k]+a[k]\left(∆t\right)+\frac{{∆t}^{2}}{2}w[k]$$

The equation states that the new velocity at time step *k* + 1 is determined by the current velocity, the acceleration multiplied by the time step, and a noise term:(21)$$a[k+1]=a[k]+\left(∆t\right)w[k]$$

The equation states that the new acceleration at time step *k* + 1 is determined by the current acceleration and a noise term.

The choice of noise term *w*$\left[k\right]$ depends on the characteristics of the system and the sources of uncertainty or disturbances. As mentioned earlier, it is common to assume $w\left[k\right]$ to be white noise. White noise has equal power at all frequencies, making it a simple and convenient choice in many applications.

The state equation can be represented in matrix form, as follows:(22)X[k+1]=AX[k]+Gw[k]

The matrices *A* and *G* represent the transition matrix and noise process matrix, respectively. Process noise vector and *Q* covariance can be defined as follows [16,17,18]:$$w[k]=[w_{x}[k];\;w_{y}[k];\;w_{z}[k]],\;Q=diag([\sigma_{ax}^{2}\;\sigma_{ay}^{2}\;\sigma_{az}^{2}])$$
(23)$$A=\left[\begin{array}{ccccccccc} 1 & 0 & 0 & ∆t & 0 & 0 & {∆t}^{2}/2 & 0 & 0\\ 0 & 1 & 0 & 0 & ∆t & 1 & 0 & {∆t}^{2}/2 & 0\\ 0 & 0 & 1 & 0 & 0 & ∆t & 0 & 0 & {∆t}^{2}/2\\ 0 & 0 & 0 & 1 & 0 & 0 & ∆t & 0 & 0\\ 0 & 0 & 0 & 0 & 1 & 0 & 0 & ∆t & 0\\ 0 & 0 & 0 & 0 & 0 & 1 & 0 & 0 & ∆t\\ 0 & 0 & 0 & 0 & 0 & 0 & 1 & 0 & 0\\ 0 & 0 & 0 & 0 & 0 & 0 & 0 & 1 & 0\\ 0 & 0 & 0 & 0 & 0 & 0 & 0 & 0 & 1 \end{array}\right]\;,G=\left[\begin{array}{ccc} \frac{{∆t}^{3}}{6} & 0 & 0\\ 0 & \frac{{∆t}^{3}}{6} & 0\\ 0 & 0 & \frac{{∆t}^{3}}{6}\\ \frac{{∆t}^{2}}{2} & 0 & 0\\ 0 & \frac{{∆t}^{2}}{2} & 0\\ 0 & 0 & \frac{{∆t}^{2}}{2}\\ ∆t & 0 & 0\\ 0 & ∆t & 0\\ 0 & 0 & ∆t \end{array}\right]$$

The observation vector, *o*$\left[k\right],$ contains the distance values used in the geometric approach and an additional noise vector. Since the observation vector typically contains noisy or imperfect measurements of the true state of the system, the measurement vector, *H*[*k*], represents the predicted measurements based on the current estimate of the system’s state. The observation vector contains the real-world measurements obtained from sensors, while the measurement vector represents the predicted measurements based on a current estimate of the system’s state:(24)$$o[k]=\left[\begin{array}{c} d_{1}\left[k\right]+n_{1}\left[k\right]\\ d_{2}\left[k\right]+n_{2}\left[k\right]\\ d_{3}\left[k\right]+n_{3}\left[k\right]\\ d_{4}\left[k\right]+n_{4}\left[k\right] \end{array}\right]\approx H[k]x[k]+n[k]$$
$d_{1}$, …, $d_{4}$ represents the true distance measurements. In addition to this, *n*_1_, *n*_2_, *n*_3_, and *n*_4_ represent the noise associated with each measurement.

The measurement vector is represented by *H*[k], and the vector of noise, with a mean of zero and a covariance matrix, is represented by *n*$\left[k\right].$ As seen in Figure 3, in the general structure, five Marvelmind ultrasonic sensors are used for the system, with one being mobile and four being stationary. The distances from the four stationary sensors to the mobile sensor can be defined as *r*_1_, *r*_2_, *r*_3_, and *r*_4_. The matrix, *R*, a covariance matrix, is defined by the covariances of these distance data as follows:(25)$$R=diag([\sigma_{r1}^{2}\;\sigma_{r2}^{2}\;\sigma_{r3}^{2}\;\sigma_{r4}^{2}])$$

The equations described in the structure of the applied EKF are utilized. The initialization stage of this EKF is one of the crucial factors, where the initial values of the covariance matrices are assigned. Subsequently, the prediction and update steps are executed in sequence [15,19]. (1)Prediction of state:(26)$$\bar{x}[k]=f(A\hat{x}\left[k-1],u[k]\right)$$
where f is a nonlinear transition function and *u*[*k*] represents control inputs if applicable.
(2)Prediction of state covariance:(27)$$\bar{P}[k]=A\hat{P}[k-1]A^{T}+Q$$
where *A_k_*−_1_ is the Jacobian matrix of the nonlinear transition function, *f*, with respect to the state, X.
(3)Gain calculation of the EKF:(28)$$K[k]=\bar{P}[k]{H\left[k\right]}^{T}{\left[H\left[k\right]\bar{P}\left[k\right]{H\left[k\right]}^{T}+R[k]\right]}^{-1}$$
(4)State correction:(29)$$\hat{x}[k]=\bar{x}[k]+K[k](O[k]-h(\;\bar{x}[k]))$$
where *h* is the Jacobian matrix and represents the nonlinear measurement function.
(5)State covariance correction:(30)$$\hat{P}[k]=(I-K[k]H[k])\bar{P}[k]$$

In EKF, the complexity of computation is increased by calculating the Jacobian matrix. To enhance the efficiency and accuracy, and to decrease the computational cost of the EKF computations, UDU factorization is applied. The details of this algorithm are given in the next section.

### 2.3. UDU Factorization

UDU factorization employs the above equations. It substitutes the covariance matrix, *P*[*k*], with an upper triangular matrix with ones on the diagonal (*U*[*k*]) and a diagonal matrix, *D*[*k*], and can be expressed as follows [20]:(31)$$P[k]=U[k]D[k]U^{T}[k]$$

### 2.4. UDU Measurement Update

Reword the sentence by expressing the process of factorizing the covariance, *P,* into an *LDL* form, where instead of utilizing an upper triangular matrix a lower triangular matrix is used [21].
(32)$$P[k]=L[k]∆[k]{L[k]}^{T}\;\mathrm{and}\;\bar{P}[k]=\bar{L}[k]\bar{∆}[k]{\bar{L}[k]}^{T}$$
where ∆ is the diagonal matrix and *U* and *D* are the inverses of *L^T^* and ∆, respectively [21].
(33)$$\begin{array}{c} {P[k]}^{-1}={L[k]}^{-T}{∆[k]}^{-1}{L[k]}^{-1\;}=U\left[k\right]D[k]{U[k]}^{T}\\ {\bar{P}[k]}^{-1}={\bar{L}[k]}^{-T}{\bar{∆}[k]}^{-1}{\bar{L}[k]}^{-1\;}=\bar{U}\left[k\right]\bar{D}[k]{\bar{U}[k]}^{T} \end{array}$$

Therefore, the measurement update becomes as follows:(34)$$U\left[k\right]D[k]{U\left[k\right]}^{T}=\bar{U}[k]\bar{D}[k]{\bar{U}\left[k\right]}^{T}+{H\left[k\right]}^{T}{R\left[k\right]}^{-1}H[k]$$

Then, factorize the *mxm* matrix *R_k_* into the *LDL* form as described in [21]:(35)$$\begin{array}{c} R[k]=L_{R[k]}{∆}_{R[k]}L_{R[k]}^{T}\\ {R[k]}^{-1}=L_{R[k]}^{-T}{∆}_{R[k]}^{-1}L_{R[k]}^{-T}=U_{R[k]}D_{R[k]}U_{R[k]}^{T} \end{array}$$

Therefore, (34) becomes as follows:(36)$$U\left[k\right]D[k]{U\left[k\right]}^{T}=\bar{U}[k]\bar{D}[k]{\bar{U}\left[k\right]}^{T}+{H\left[k\right]}^{T}U_{R\left[k\right]}D_{R\left[k\right]}U_{R\left[k\right]}^{T}H[k]$$
and the term $U_{R[k]}H$ can be expressed as follows [21]:(37)$$U_{R[k]}^{T}H_{R}=\left[\begin{array}{c} v_{1}^{T}\\ \vdots \\ v_{m}^{T} \end{array}\right]$$

Each v_i_ is an *n* × 1 vector. The factor ${H[k]}^{T}{R[k]}^{-1}H[k]$ can be expressed as follows [21]:(38)HkTRk−1Hk=HkTURkDRkURkTHk=v1T⋮vmTT1d1R⋯0⋮⋱⋮0⋯1dmRv1T⋮vmT=∑i=1m1diRviviT

Therefore, the measurement update is expressed as follows:(39)$$U\left[k\right]D[k]{U[k]}^{T}=\bar{U}[k]\bar{D}[k]{\bar{U}[k]}^{T}+\sum_{i=1}^{m}\frac{1}{d_{i_{R}}}v_{i}v_{i}^{T}$$

In addition to this, $\hat{z}\left[k\right]$ and $\bar{z}[k]$, which are related to ${\hat{x}}_{k}$ and ${\bar{x}}_{k}$, are defined below [20,21]:(40)$$\begin{array}{c} \hat{z}[k]≜{P\left[k\right]}^{-1}\hat{x}[k],\;\bar{z}≜{\bar{P}\left[k\right]}^{-1}\bar{x}[k]\\ \hat{z}[k]={P\left[k\right]}^{-1}\left(I-K\left[k\right]H\left[k\right]\right)\bar{x}[k]+{P\left[k\right]}^{-1}K[k]m[k]\\ \hat{z}[k]=\bar{z}[k]+{H\left[k\right]}^{T}{R\left[k\right]}^{-1}m[k] \end{array}$$
where m[k] is the measurement vector.

### 2.5. UDU Time Update

The Kalman filter uses a standard method. It involves inverting the information matrix to obtain the covariance, then propagating the covariance, and finally inverting the propagated covariance matrix to prepare for the measurement update step. This proposed algorithm utilizes this method for covariance propagation and factorizing *Q*[*k*] with UDU parameterization [20,21]:(41)$$Q[k]=U_{Q[k]}{∆}_{Q[k]}U_{Q[k]}^{T}$$
where ${∆}_{Q[k]}$ is a diagonal *p* × *p* matrix, and $G_{Q[k]}$ is defined as follows:(42)$$G_{Q[k]}≜G[k]U_{Q[k]}$$
so $\bar{P}[k]$ becomes the following:(43)$$\bar{P}[k+1]=\phi [k]P[k]{\phi [k]}^{T}+G_{Q[k]}{∆}_{Q[k]}G_{Q[k]}^{T}$$

Using the matrix inversion lemma, we obtain the following:$${\left(Q+WEF\right)}^{-1}=Q^{-1}-Q^{-1}W{\left(E^{-1}+FQ^{-1}W\right)}^{-1}FQ^{-1}$$
(44)$$\begin{array}{c} Q=\phi \left[k\right]P[k]{\phi [k]}^{T};\;E={∆}_{Q[k]};\;W=G_{Q[k]};\;F=G_{Q[k]}^{T}\\ {\;Q}^{-1}=M[k]{\phi [k]}^{-1}{P[k]}^{-1}{\phi [k]}^{-1} \end{array}$$

The inversion of the propagated covariance matrix is as follows [20,21]:$${\bar{P}[k+1]}^{-1}=M[k]-M[k]G_{Q\left[k\right]}{\left[{G\left[k\right]}^{T}M\left[k\right]G_{Q\left[k\right]}+{∆}_{Q\left[k\right]}^{-1}\right]}^{-1}G_{Q\left[k\right]}^{T}M[k]$$
(45)$$\bar{G}[k]={\phi [k]}^{-1}G_{Q[k]},\;D_{Q[k]}={∆}_{Q[k]}^{-1}$$

Then, the expression for ${P[k+1]}^{-1}$ is given by the following:(46)$${P[k+1]}^{-1}={\phi [k]}^{-T}\left\{{P[k]}^{-1}-{P[k]}^{-1}\bar{G}[k]{\left[{\bar{G}[k]}^{T}{P[k]}^{-1}\bar{G}[k]+D_{Q[k]}\right]}^{-1}{\bar{G}[k]}^{T}{P[k]}^{-1}\right\}{\phi [k]}^{-1}$$
and the Kalman (*K_UDU_*) gain is defined as follows:(47)$$K_{UDU[k]}≜{P[k]}^{-1}\bar{G}[k]{\left[{\bar{G}[k]}^{T}{P[k]}^{-1}\bar{G}[k]+D_{Q[k]}\right]}^{-1}$$

Then, if we define brackets in (47) as ${P[k]}^{-1}$, we obtain the following:(48)$$\begin{array}{c} {P[k]}^{-1}≜{P[k]}^{-1}-{P[k]}^{-1}\bar{G}[k]{\left[{\bar{G}[k]}^{T}{P[k]}^{-1}\bar{G}[k]+D_{Q[k]}\right]}^{-1}{\bar{G}[k]}^{T}{P[k]}^{-1}\\ {P[k]}^{-1}=\left[I-K_{UDU[k]}{\bar{G}[k]}^{T}\right]{P[k]}^{-1} \end{array}$$

Equation (48) is an analog to Equations (28)–(30), with the following:(49)$${P[k]}^{-1}\to \hat{P}[k],\;{P[k]}^{-1}\to \bar{P}[k],\bar{G}\left[k\right]\to {H\left[k\right]}^{T},K_{UDU[k]}\to K\left[k\right],D_{Q[k]}\to R[k]$$
and since $D_{Q[k]}$ is the diagonal matrix, the UDU factorization of ${P[k]}^{-1}$ is solved directly by using Carlson’s rank-one update [22]:(50)$$\begin{array}{cc} Ư[k]Ɗ[k]{Ư[k]}^{T} & =U\left[k\right]D[k]{U\left[k\right]}^{T}\\ & -U[k]D[k]{U[k]}^{T}\bar{G}[k][{\bar{G}[k]}^{T}U\left[k\right]D[k]{U[k]}^{T}\bar{G}[k]\\ & +{∆}_{Q[k]}^{-1}]{\bar{G}[k]}^{T}U[k]D[k]U[k] \end{array}$$
and time-propagated UDU factors of ${\bar{P}[k+1]}^{-1}$ are defined as below:(51)$${\bar{P}[k+1]}^{-1}=\bar{U}[k+1]\bar{D}[k+1]{\bar{U}[k+1]}^{T}={\phi [k]}^{-T}Ư[k]Ɗ[k]{Ư[k]}^{T}{\phi [k]}^{-1}$$

The time update for $\bar{z}[k+1]$ is obtained as follows.
(52)$${\bar{P}[k+1]}^{-1}\bar{x}[k+1]={\bar{P}\left[k+1\right]}^{-1}\phi [k]P[k]{P\left[k\right]}^{-1}\hat{x}[k]$$
then it becomes
(53)$$\bar{z}[k+1]={\bar{P}\left[k+1\right]}^{-1}\phi [k]P[k]\hat{z}[k]$$
and it can be written as:(54)$$\bar{z}[k+1]={\phi [k]}^{-T}\left[I-K_{UDU[k]}{\bar{G}[k]}^{T}\right]\hat{z}[k]$$

The equation systems specified in this section have been utilized to enhance the performance of the EKF structure in real-time applications. The outcomes of this structure will be comprehensively analyzed in Section 4 of the article. The next section will explain how data are collected from the Marvelmind sensor.

## 3. Data Acquisition

This section will provide detail on how to retrieve data from the Marvelmind sensor. The mobile Marvelmind sensor is connected to a laptop computer via the serial port. To ensure that all of the necessary components are working correctly, four Marvelmind sensors attached to the wall were activated, and their functionality was confirmed through the Marvelmind’s own interface [23]. After verifying that all of the sensors were operating properly, a software tool on the ROS platform was run on the laptop to gather the data shown in Table 1.

After the data were collected in the ROS environment, the packet parsing functions that interpret the data were added to Pixhawk in order to be able to collect them in the same way from Pixhawk as well. In addition to the written codes, a physical connection must also be made between Pixhawk and the Marvelmind sensor. This physical connection schema can be seen in Figure 4 [24].

Following the schematic connection depicted in Figure 5a,b, a physical connection between Pixhawk and the Marvelmind sensor was established. Data collection was then conducted on Pixhawk, while the other Marvelmind sensors remained fixed to the wall.

The explanation of the algorithm tests and their outcomes will be provided in detail in the next section.

## 4. Testing Algorithm and Results

The UDU-factorized EKF structure and the geometric approach based on the RLS method were implemented in the C++ environment using MATLAB code generation support. The integration was carried out on the Pixhawk platform. The trajectories obtained from the Marvelmind sensor are presented in Table 2. The robotic platform used in this study is a quadcopter. Due to reliability concerns associated with indoor flight, the quadcopter was constrained to a fixed table, and data were collected using Marvelmind sensors. In the initial six trajectories, no motion was applied to the robot along the z-axis; however, intentional movement in the z-axis was introduced during the data collection for the last two trajectories to assess the algorithm’s performance.

The position estimations, calculated solely through the trilateration algorithm, and the positions derived by integrating the trilateration with the EKF and UDU factorization algorithms, incorporating IMU data, are presented in Figure 6, Figure 7, Figure 8, Figure 9, Figure 10, Figure 11, Figure 12 and Figure 13. Additionally, the locations of the fixed sensors used in these figures are indicated as black dots. Comprehensive information on the minimum, maximum, and average error values for the calculated positions is provided in Table 3, generated using the RMS value.

It is evident from the figures that there are differences between the position estimations obtained using UDU factorization applied to the EKF structure and those obtained solely through the geometric approach. The position estimation derived from UDU factorization is closer to the reference position compared to the geometric approach. This is expected because the geometric approach calculates position estimations solely based on distance data from the Marvelmind sensor. On the other hand, in the approach utilizing the EKF, a sensor fusion algorithm is employed, incorporating both distance and accelerometer data to enhance the accuracy of position estimations.

In Figure 6, Figure 7 and Figure 8, four mobile and one fixed Marvelmind sensors were utilized. In Figure 9, Figure 10 and Figure 11, three mobile and one fixed Marvelmind sensors were used. In Figure 12 and Figure 13, both four mobile and one fixed in addition to three mobile and one fixed Marvelmind sensors were sequentially employed. In the first six scenarios, there was no movement along the *z*-axis. Conversely, in the last two scenarios, motion along the *z*-axis was introduced. At this point, the obtained RMS values exhibit differences depending on the trajectories, the number of sensors used, and the algorithmic structure.

When examining Table 3, it is apparent that the results obtained solely through the geometric approach for all trajectories fall short of the desired accuracy, with a notably high error amount; however, it is observed that position estimations reached the desired levels when UDU-EKF was employed in conjunction with the geometric approach. The primary reason for this is, as detailed in the article, that the sensor fusion algorithm is capable of providing a more accurate estimation compared to the geometric-based algorithm. Incorporating accelerometer data into the position estimation process contributes to achieving a more accurate outcome.

For each trajectory, the geometric approach, relying solely on trilateration, exhibits notable variability in accuracy across trajectories, with maximum errors reaching up to 5.4721 m in Trajectory 6. In contrast, the integration of UDU factorization into the EKF consistently enhances accuracy, as demonstrated by reduced minimum, mean, and maximum errors. Trajectory 6 stands out, with a remarkable improvement, showcasing a substantial drop in the maximum error from 5.4721 to 0.4796 m. This underscores the pivotal role of UDU-EKF in refining position estimates, particularly in scenarios involving intentional *z*-axis movement. The overall pattern highlights the algorithm’s adaptability and robustness in dynamic indoor environments, reaffirming its efficacy for precise positioning in real-world applications.

In addition to that, the code structure was running multiple times for the comparison of UDU factorization EKF with normal EKF in terms of processing speeds, and the processing times for both structures were observed. The comparison of processing speeds between the EKF with UDU factorization and the conventional EKF, as detailed in Table 4, offers valuable insights into their computational efficiency. The experimental setup involved multiple runs to ensure a thorough assessment, and the recorded processing times for UDU-EKF and the EKF, presented in seconds for each trajectory, quantify the time required for algorithm execution. The observed results confirm the consistent outperformance of the EKF structure with UDU factorization, with a minimum increase in the processing time of 18%. The percentage difference analysis underscores the substantial improvement in processing speed achieved by incorporating UDU factorization. Trajectory-specific performance evaluations reveal notable percentage differences of 21%, 25%, and 18% for Trajectories 1, 2, and 3, respectively, emphasizing UDU-EKF’s faster execution across diverse scenarios. The implications for real-world applications are substantial, especially in time-sensitive scenarios where quick and accurate positioning is paramount. The observed reduction in computational cost further positions UDU factorization as an efficient choice, particularly for resource-constrained systems. These findings suggest that implementing UDU factorization in the EKF structure can be considered a strategic optimization for improving overall algorithm efficiency. In summary, the analysis highlights UDU-EKF’s superiority in processing speed, showcasing its potential to significantly enhance the efficiency of positioning algorithms in critical real-world applications.

As a result, the integration of UDU factorization into the EKF structure has been observed to improve the accuracy of position estimation, as evidenced by both the graphs and Table 3. UDU-EKF achieves reliable results even in scenarios involving motion along the three axes. Additionally, upon reviewing Table 4, it is evident that the UDU-EKF structure operates faster than the traditional EKF. Considering both algorithms, it is observed that UDU-EKF plays a significant role in position estimation with lower computational cost.

## 5. Conclusions

This paper presents a comprehensive study on the integration of UDU factorization into the structure of the EKF for indoor position estimation algorithms. The proposed algorithm demonstrates its effectiveness in real-time systems, showcasing improved accuracy in position estimation compared to traditional EKF approaches. The decomposition of the EKF structure with UDU factorization aims to reduce computational costs, making it particularly suitable for real-time applications.

The study focuses on a geometric method based on reference points, introducing an approach that utilizes distance measurements and quadratic equations to determine the position coordinates of a target point.

Experimental results, conducted using Marvelmind sensors and a quadcopter, provide a thorough analysis of the algorithm’s performance. Trajectory-based comparisons emphasize the superiority of UDU-EKF over the geometric approach, highlighting its ability to achieve more accurate position predictions. The integration of UDU factorization into the EKF not only enhances accuracy but also demonstrates adaptability to dynamic indoor environments.

Furthermore, the paper investigates the computational efficiency of the proposed algorithm, comparing processing speeds between UDU-EKF and traditional EKF structures. The results consistently show that UDU-EKF outperforms traditional EKF in terms of processing speed. This finding is crucial for real-world applications, especially those requiring quick and accurate positioning.

In summary, the integration of UDU factorization into the EKF structure emerges as a strategic optimization for achieving higher accuracy with lower computational costs. The algorithm’s adaptability to all three axes and its superior processing speed position it as a promising solution for real-time positioning systems, emphasizing its potential impact on various applications, including robotics and indoor navigation.

## Figures and Tables

**Figure 1 sensors-24-03048-f001:**
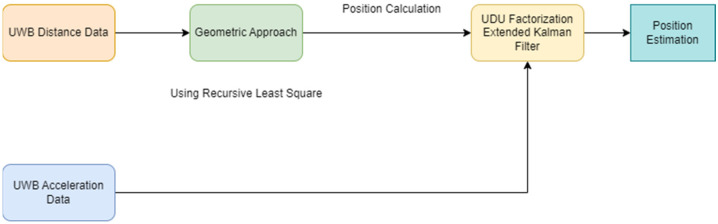
General structure of position estimation.

**Figure 2 sensors-24-03048-f002:**
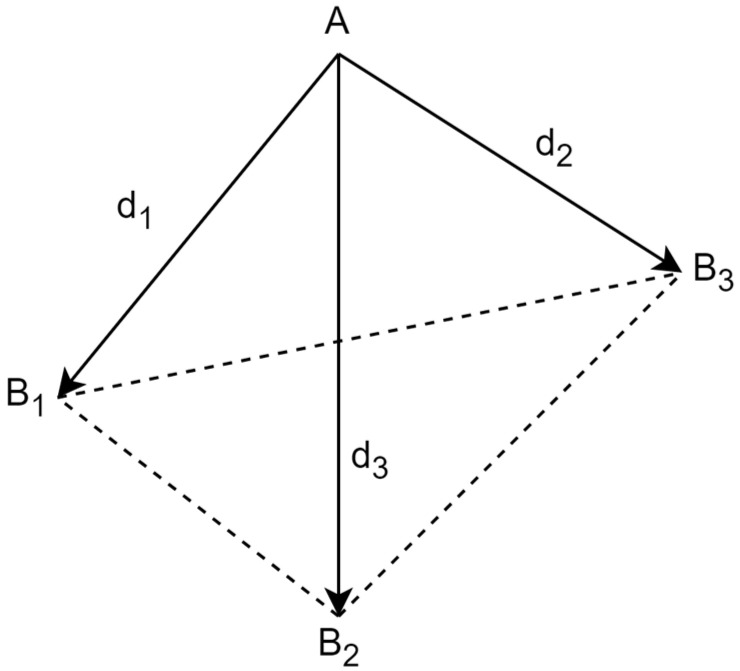
Reference points and interval measurements.

**Figure 3 sensors-24-03048-f003:**
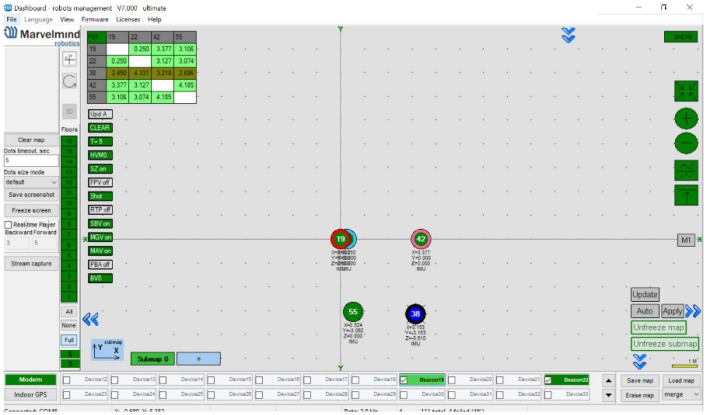
Displaying used sensors on the interface.

**Figure 4 sensors-24-03048-f004:**
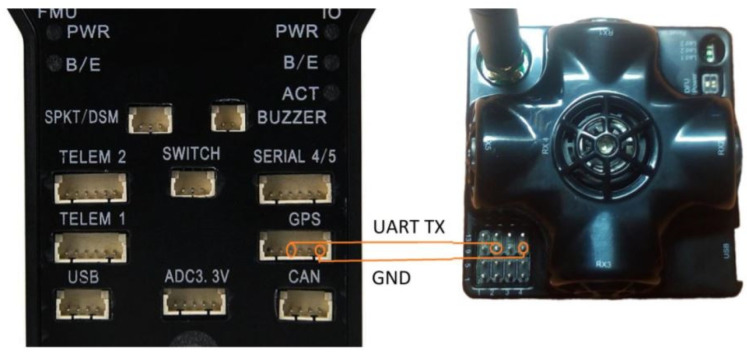
Pixhawk physical wiring diagram with the Marvelmind sensor [24].

**Figure 5 sensors-24-03048-f005:**
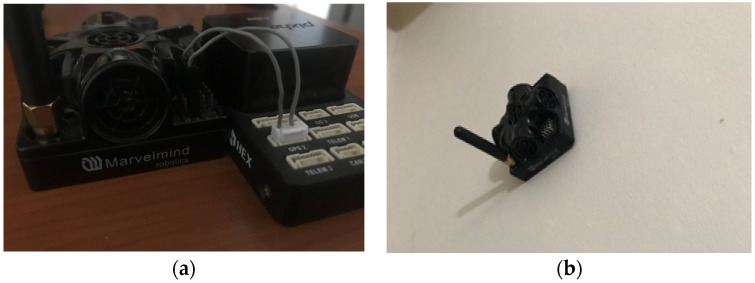
(**a**). Pixhawk physical connection with the Marvelmind sensor (**b**). Marvelmind sensor fixed on the wall.

**Figure 6 sensors-24-03048-f006:**
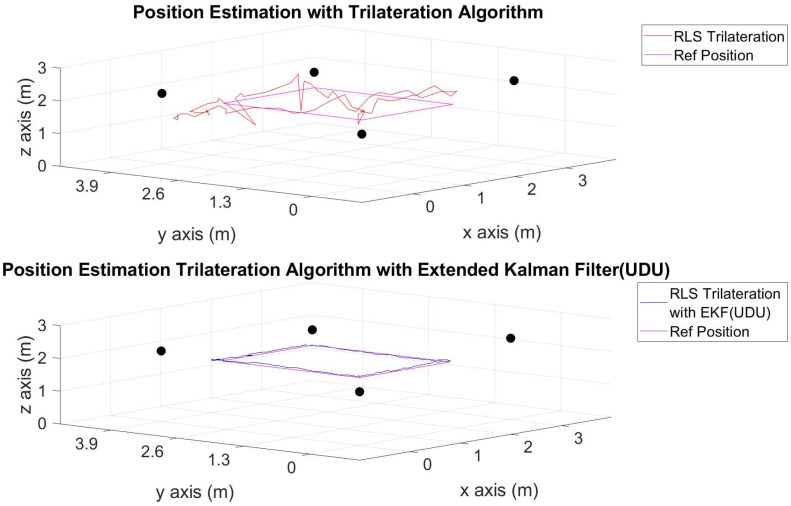
Position-estimation-only trilateration algorithm and trilateration algorithm with the EKF in Trajectory 1.

**Figure 7 sensors-24-03048-f007:**
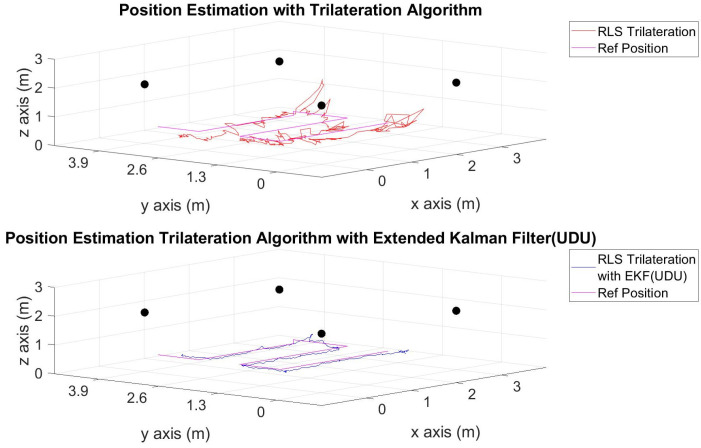
Position-estimation-only trilateration algorithm and trilateration algorithm with the EKF in Trajectory 2.

**Figure 8 sensors-24-03048-f008:**
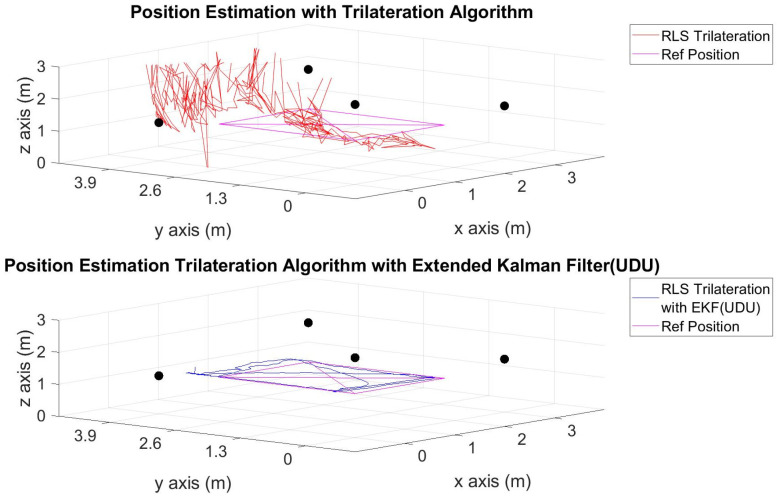
Position-estimation-only trilateration algorithm and trilateration algorithm with the EKF in Trajectory 3.

**Figure 9 sensors-24-03048-f009:**
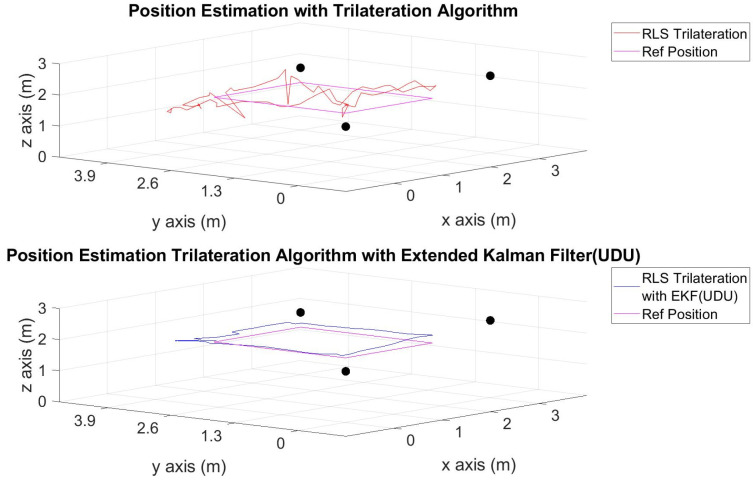
Position-estimation-only trilateration algorithm and trilateration algorithm with the EKF in Trajectory 4.

**Figure 10 sensors-24-03048-f010:**
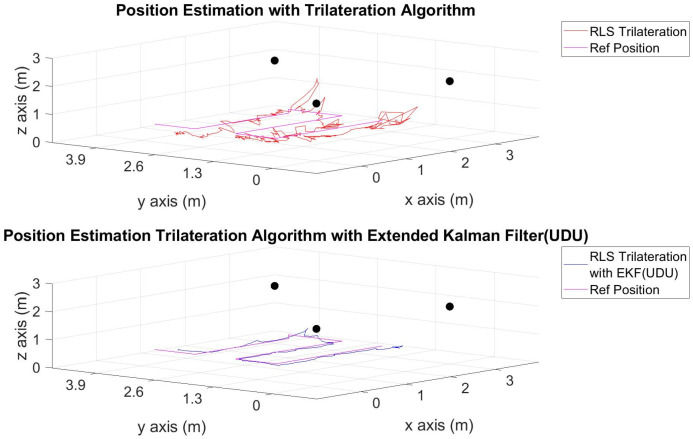
Position-estimation-only trilateration algorithm and trilateration algorithm with the EKF in Trajectory 5.

**Figure 11 sensors-24-03048-f011:**
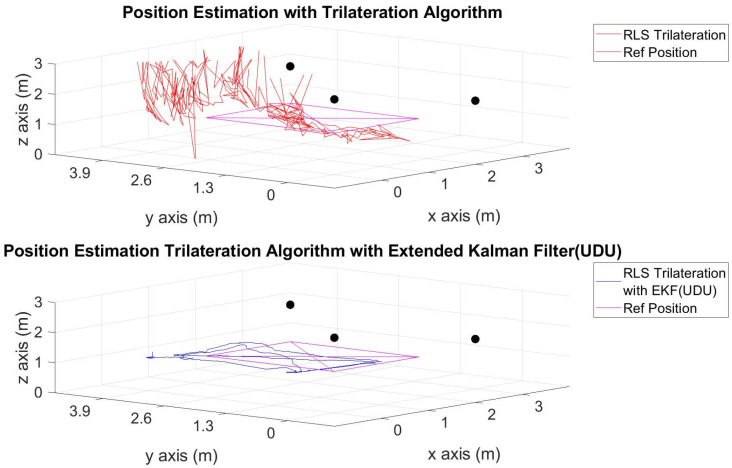
Position-estimation-only trilateration algorithm and trilateration algorithm with the EKF in Trajectory 6.

**Figure 12 sensors-24-03048-f012:**
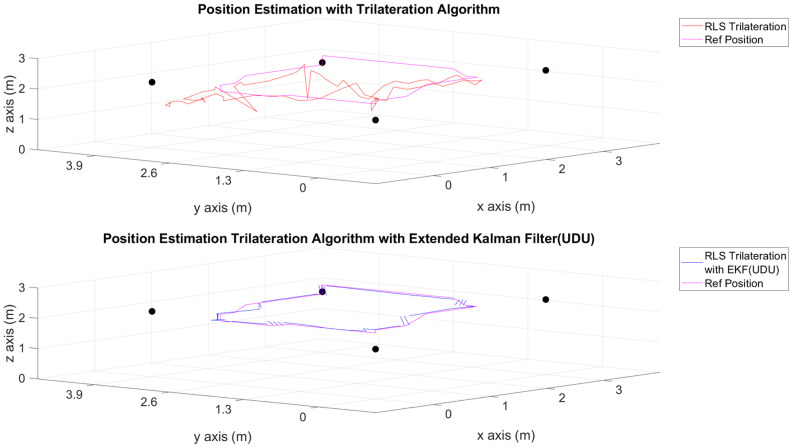
Position-estimation-only trilateration algorithm and trilateration algorithm with the EKF in Trajectory 7.

**Figure 13 sensors-24-03048-f013:**
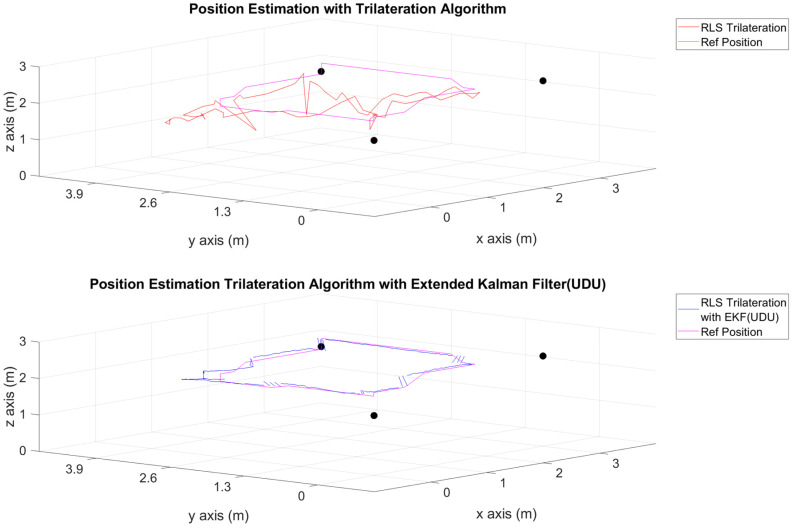
Position-estimation-only trilateration algorithm and trilateration algorithm with the EKF in Trajectory 8.

**Table 1 sensors-24-03048-t001:** Marvelmind sensor data structure [23].

Message	Message Variable	Data Type	Description of Data
beacon_distance	dist_m	float64	Raw distance data of beacon, in meters
	add_hedge	uint8	Address no of mobile beacon
	add_beacon	uint8	Address no of stationary beacon
hedge_imu_raw	time_var	int64	Timestamp of IMU data
	accel_x	int16	Accelerometer of x data
	accel_y	int16	Accelerometer of y data
	accel_z	int16	Accelerometer of z data

**Table 2 sensors-24-03048-t002:** Trajectories of the tests.

Trajectory ID	Beacon Number	Trajectories
1	5 (4 stationary + 1 mobile)	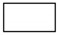
2	5 (4 stationary + 1 mobile)	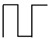
3	5 (4 stationary + 1 mobile)	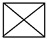
4	4 (3 stationary + 1 mobile)	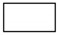
5	4 (3 stationary + 1 mobile)	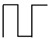
6	4 (3 stationary + 1 mobile)	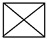
7	5 (4 stationary + 1 mobile)	With a different z value 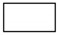
8	4 (3 stationary + 1 mobile)	With a different z value 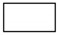

**Table 3 sensors-24-03048-t003:** Error tables.

Trajectory ID	Algorithm	Min Error (m)	Mean Error (m)	Max Error (m)
1	Only geometric approach	0.0734	0.3356	0.9940
1	Geometric approach with the EKF (UDU)	0.0272	0.0617	0.1439
2	Only geometric approach	0.0776	0.4533	1.7680
2	Geometric approach with the EKF (UDU)	0.0177	0.0827	0.1686
3	Only geometric approach	0.3536	1.2583	5.0031
3	Geometric approach with the EKF (UDU)	0.0191	0.1034	0.2148
4	Only geometric approach	0.1142	0.3628	1.1472
4	Geometric approach with the EKF (UDU)	0.0430	0.2070	0.4387
5	Only geometric approach	0.1348	0.5312	2.1322
5	Geometric approach with the EKF (UDU)	0.0752	0.1660	0.3361
6	Only geometric approach	0.4612	1.436	5.4721
6	Geometric approach with the EKF (UDU)	0.1141	0.2378	0.4796
7	Only geometric approach	0.0682	0.3781	1.2343
7	Geometric approach with the EKF (UDU)	0.0305	0.1312	0.1980
8	Only geometric approach	0.0836	0.4116	1.3841
8	Geometric approach with the EKF (UDU)	0.0504	0.2162	0.3226

**Table 4 sensors-24-03048-t004:** Process time difference of the EKF vs. UDU-EKF.

Trajectory	UDU-EKF (s)	EKF (s)	Percentage of Difference
1	0.0094	0.0119	21%
2	0.0076	0.0101	25%
3	0.0099	0.0121	18%

## Data Availability

Data are contained within the work and article.

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
