# Peer review of "3D Indoor Position Estimation Based on a UDU Factorization Extended Kalman Filter Structure Using Beacon Distance and Inertial Measurement Unit Data†"

_sensors, 2024, doi:10.3390/s24103048_

Round 1

Reviewer 1 Report

Comments and Suggestions for Authors

This paper proposes a 3-D position estimation algorithm which combines the data from ultrasonic sensors and IMU utilizing the EKF with UDU factorization. The RLS method is used to estimate the position with raw distance data from the ultrasonic sensors and combined with IMU data to provide a position solution similar to GPS.The designed algorithm was implemented on the Pixhawk development board and run in real time. Test results have shown that the UDU-EKF structure integrated into the embedded system is faster than the classical EKF. It can enlighten for the researchers in the field of indoor position. However, there are some problems as below,

 1.In the experimental section, the comparison results of various algorithms can be further discussed, and their specific characteristics can be listed. At the same time, in order to reflect the 3D positioning effect, a three-dimensional flight trajectory can be set, not just in the form of a planar trajectory.

 2.The proposed algorithm can be illustrated in a diagram to aid in expression. At the same time, for the proposed method, focus on elaborating on special or important areas that need to be considered, rather than comprehensively listing the formula model.

Comments on the Quality of English Language

no comments

Author Response

Thank you very much for the constructive review which resulted in a better and stronger manuscript. Based on your comments, I made the following changes in the article:

1.In the comparison section on the experimental side, I provided a more detailed explanation of the comparison results of the algorithms. Additionally, as you suggested, I collected a dataset by creating a variable route in the z-axis, and added it to the results section. When the algorithm is run with the collected dataset, it can be observed from the relevant graph and table, that it compensates for changes in the z-axis. This indicates that the algorithm provides good results not only in the planar trajectory but also in all three axes.

2.I added a diagram explaining the basic structure of the algorithm. I also provided a more concrete explanation of why the UDU EKF structure is used. By reducing the matrix calculation time, I believe this will pave the way for the algorithm to be used with other modules in the future. Since the frequencies of modules in real-time systems are crucial, the significance of this study becomes even more apparent. Accelerating the matrix calculations within the module will enable the indoor position calculation module to be used at the desired frequency. Moreover, with better optimization, it can be made to operate at even higher speeds. This will depend on the system to be designed.

Reviewer 2 Report

Comments and Suggestions for Authors

For this paper, I have some confusions.

1.       Kalman Filter is interesting and old topic, which is used for Indoor Location Positioning. Do you consider the topic original or relevant in the field? Does it address a specific gap in the field?

2.       I think the innovation of this article needs to be improved. After reading, I do not know your research improvement. What is the main question you tackled in your work?

3.       In the Table, I think you’d better indicate your own practices so that your work can be displayed. You can also illustrate some comments for the table or figures.

4.       When compared with other algorithms, I think you’d better add the reference for the algorithm so that it is clear for the reader.

5.       There are many grammar mistakes and nonstandard formula writing.

Like the description in Introduction

Therefore, an algorithm is designed to performs more accurate indoor positioning 55 estimation.

And the description “As seen Figure 2,” I think it should be “As seen in Figure 2,”

I think the sentence should be corrected.

6.       In Equation (21), it presents a[k+1] = a[k]+ (∆t)w[k]. I am curious about why it does not contain the noise part. Please explain it.

7.       Writing norms need to be noted. The variables before and after are not italicized. Does italicization at the end of the article represent different variables? If different, please use different variables, otherwise this can easily cause misunderstandings.

Comments on the Quality of English Language

For this paper, I have some confusions.

1.       Kalman Filter is interesting and old topic, which is used for Indoor Location Positioning. Do you consider the topic original or relevant in the field? Does it address a specific gap in the field?

2.       I think the innovation of this article needs to be improved. After reading, I do not know your research improvement. What is the main question you tackled in your work?

3.       In the Table, I think you’d better indicate your own practices so that your work can be displayed. You can also illustrate some comments for the table or figures.

4.       When compared with other algorithms, I think you’d better add the reference for the algorithm so that it is clear for the reader.

5.       There are many grammar mistakes and nonstandard formula writing.

Like the description in Introduction

Therefore, an algorithm is designed to performs more accurate indoor positioning 55 estimation.

And the description “As seen Figure 2,” I think it should be “As seen in Figure 2,”

I think the sentence should be corrected.

6.       In Equation (21), it presents a[k+1] = a[k]+ (∆t)w[k]. I am curious about why it does not contain the noise part. Please explain it.

7.       Writing norms need to be noted. The variables before and after are not italicized. Does italicization at the end of the article represent different variables? If different, please use different variables, otherwise this can easily cause misunderstandings.

Author Response

Thank you very much for the constructive review which resulted in a better and stronger manuscript. Based on your comments, I made the following changes in the article:

1)The Kalman filter, as you mentioned, is an old and widely used method. In indoor positioning, the EKF structure is utilized in various systems. However, integrating the UDU structure to speed up the matrix operations of the Kalman filter, specifically for indoor positioning, is a novel approach that hasn't been used before, at least not in the literature. Additionally, I believe that by creating my own solution set using distance data and accelerometer data and supporting it with UDU-EKF, I have designed a more accurate and faster position system. When all of these are considered, it is observed that the designed algorithm will fill a gap in indoor position estimation

2) In the scope of my study, I developed a system that performs indoor positioning based on the working principle of GPS. In designing this, I opted for a real-time system structure using UDU factorization in the EKF (Extended Kalman Filter) instead of the classical EKF structure. This allowed for faster computations in covariance calculations of the Kalman filter by utilizing only the upper diagonal part. The system I designed is capable of delivering efficient performance in real-time systems that require swift operation. In summary, addressing the question of how to perform indoor location estimation both more accurately and faster, I conducted this study.

3) For the figures and tables in the test section, I integrated more detailed explanations into the article. As you suggested, I believe that by using more explanatory concepts, I have made it easier for the reader to understand.

4) Actually, for both the geometric structure, EKF, and UDU factorization, I made references within the text. You can see these below and in the references section as well

[11] Bodrumlu T, Caliskan F. “Indoor Position Estimation Using Ultrasonic Beacon Sensors and Extended Kalman Filter”. Engineering Proceedings. 2022; 27(1):16.

[12] Norrdine, A. An Algebraic Solution to the Multilateration Problem. In Proceedings of the 2012 International Conference on Indoor Positioning and Indoor Navigation, Sydney, Australia, 13–15 November 2012; pp. 1–4

[16] Benini, A. Mancini, A. Marinelli, S. Longhi “A Biased Extended Kalman Filter for Indoor Localization of a Mobile Agent using Low-Cost IMU and UWB Wireless Sensor Network” 10th IFAC Symposium on Robot Control International Federation of Automatic Control September 5-7, 2012. Dubrovnik, Croatia

[20] C. D'souza and R. Zanetti, "Information Formulation of the UDU Kalman Filter," in IEEE Transactions on Aerospace and Electronic Systems, vol. 55, no. 1, pp. 493-498, Feb. 2019, doi: 10.1109/TAES.2018.2850379.

5) I have reviewed the grammar within the article and made the necessary corrections.

6) I did not fully understand this question. If you're asking why there is no noise, the noise term w(t) is already added within the equations, and I have explained this in the article. If you're asking why there is a noise term, it's because the accelerometer data which I collect is not perfect; it comes with noise due to external factors. I added white noise to the equation to model this.

“The choice of noise term w[k]  depends on the characteristics of the system and the sources of uncertainty or disturbances. As mentioned earlier, it is common to assume w[k]  to be a white noise. White noise has equal power at all frequencies, making it a simple and convenient choice in many applications.”

7) As you mentioned, I corrected the representations of the formulas in the article and italicized them.

Round 2

Reviewer 1 Report

Comments and Suggestions for Authors

The paper has met the requirements for revision.

Comments on the Quality of English Language

English expression meets the requirements.

Reviewer 2 Report

Comments and Suggestions for Authors

The paper has been well revised. And I think it can be published in the journal. 

Comments on the Quality of English Language

The quality of English can be improved.